# Conducting large, repeated, multi-game economic experiments using mobile platforms

Zhi Li[1,2☯], Po-Hsuan Lin[3,4☯], Si-Yuan Kong[4], Dongwu Wang[4], John Duffy[5]*

1 Department of Public Finance, School of Economics & the Wang Yanan Institute for Studies in Economics (WISE), Xiamen University, Xiamen, China, 2 The MOE Key Laboratory of Econometrics, Xiamen University, Xiamen, China, 3 Division of the Humanities and Social Sciences, California Institute of Technology, Pasadena, CA, United States of America, 4 MobLab Inc., Pasadena, CA, United States of America, 5 Department of Economics, University of California, Irvine, Irvine, CA, United States of America

☯ These authors contributed equally to this work.
* duffy@uci.edu

**Data Availability Statement:** The experimental data and code for all analyses are uploaded to Open Science Framework (OSF) and publicly accessible via DOI 10.17605/OSF.IO/KUXEN or https://osf.io/kuxen/.

## Abstract

We demonstrate the possibility of conducting synchronous, repeated, multi-game economic decision-making experiments with hundreds of subjects in-person or remotely with live streaming using entirely mobile platforms. Our experiment provides important proof-of-concept that such experiments are not only possible, but yield recognizable results as well as new insights, blurring the line between laboratory and field experiments. Specifically, our findings from 8 different experimental economics games and tasks replicate existing results from traditional laboratory experiments despite the fact that subjects play those games/task in a specific order and regardless of whether the experiment was conducted in person or remotely. We further leverage our large subject population to study the effect of large ($N = 100$) versus small ($N = 10$) group sizes on behavior in three of the scalable games that we study. While our results are largely consistent with existing findings for small groups, increases in group size are shown to matter for the robustness of those findings.

## Introduction

Economic decision-making and game theory experiments have traditionally been conducted with small numbers of subjects in laboratory settings. However, recent advances in cloud-based platforms and mobile payments mean that laboratory-type studies can now be conducted outside of the traditional laboratory setting, involving much larger numbers of participants thereby blurring the line between laboratory and field experiments. In this paper, we report on a break-through, incentivized experimental study comparing the behavior of subjects playing eight classic interactive laboratory games or individual decision-making tasks either in-person or remotely using entirely mobile devices. Our experiment took place during a summer camp at Xiamen University in July 2019 (Experiment 1, 633 subjects). It was repeated again with different Xiamen summer school participants a year later, in July 2020

**Funding:** Z.L. was supported by the National Natural Science Foundation of China (Grant No. 71873116) and the NSFC Basic Science Center Program (Grant No. 71988101). S.K. and D.W. were employed by and received salary from MobLab Inc. (www.moblab.com). J.D. is a Scientific Advisor to MobLab, a position with no compensation but with a small equity stake. Experiment 1 in 2019 was funded by MobLab Inc. Experiment 2 in 2020 was funded by Xiamen University. The funder provided support in the form of salaries for authors S.K., D.W., and J.D. but did not have any additional role in the study design, data collection and analysis, decision to publish, or preparation of the manuscript. The specific roles of these authors are articulated in the "author contributions" section.

**Competing interests:** Z.L. has no competing interest. P.L. was employed by MobLab from December 2017 to December 2018 and compensated more than US $10,000 during the last 3 years. S.K. and D.W. are employed by MobLab and compensated more than US $10,000 during the last 3 years. J.D. is a Scientific Advisor to MobLab, a position with no compensation but with a small equity stake. This does not alter our adherence to PLOS ONE policies on sharing data and materials.

(Experiment 2, 585 subjects), when subjects played remotely from home during the COVID-19 pandemic.

In both sessions, subjects used their own smartphones, tablets, or laptops, and played the same set of games using the same cloud-based platform, MobLab. They received payments based on their choices via Alipay, the mobile payment platform of Alibaba.

We have several aims in conducting and reporting on this experiment. First, we demonstrate proof-of-concept that such large scale, interactive experimentation can be done entirely using subjects' own mobile platforms to both collect data and to make payments. Second, we show that our approach yields recognizable and comparable results. Indeed our results with small groups of 2-10 subjects largely replicate findings from traditional laboratory studies despite the fact that subjects in our experiment play several different games in a sequence (a within-subjects design) as opposed to the between-subjects designs of most laboratory studies. Third, we further demonstrate that experimental results are not substantially different if we conduct our large scale experiment *in-person*, as in the 2019 Experiment 1, or *remotely with live streaming* as in the 2020 Experiment 2. These encouraging findings support the prospect of conducting efficacious and incentivized human subject experiments remotely online without sacrificing comparability to results from traditional in-person laboratory experiments. Fourth, we leverage our large subject pool to explore how group size differences of 10 (small) versus 100 (large) subjects can matter for the play of three highly scalable classic laboratory games: a *p*-beauty contest game, a voter turnout game, and a linear public goods game. We find some significant differences in behavior between the large and small groups among the three games we study for which group size is easily scalable, which reinforces the notion that group size can be an important factor in evaluating laboratory findings. In addition to the three games exploring group size effects, subjects also participated in a 2-player ultimatum bargaining game [1], an individual risk preference elicitation task similar to [2], a 2-player centipede game played 3 times [3], a 2-player trust game [4] and an individual 3-round real-effort task exploring gender differences in compensation schemes [5]. The collection of data on the play of many games by the <u>same</u> subjects enables comparisons to be made in the way subjects behave across games, which we believe will be an important area for future research using the large scale mobile experimental methodology that we propose here, and we provide some preliminary analysis in the S1 File.

Finally, we note that, to our knowledge, this study comprises two of the largest synchronous, repeated, multiple game/ decision-task experiments with paid human subjects that have ever been conducted. Our experiment is easily replicated by others as we use standard, pre-programmed games (that are easily configurable) available on the MobLab platform.

While mobile devices are now deeply embedded in people's daily lives and mobile platforms have been used to conduct field experiments, most such studies involve the collection of non-interactive, self-report data (as in RCT studies) and have been almost exclusively conducted in the domain of health research (see e.g., the survey by [6]). Exceptions include studies using mobile devices to provide subjects with nudges to complete certain tasks, e.g., an online course e.g., [7] or energy conservation, e.g., [8]. The use of subjects' own mobile devices to collect experimental data on games of strategic interaction and/or economic decision making tasks is still nascent and typically involves collection of data on a single game or task, e.g., [9]. By contrast, we consider play by subjects of a large number of games of interest to experimental researchers and show that behavior is similar to that found in traditional laboratory settings.

Other comparisons have been made between student subjects, online-workforces such as Mechanical Turk, and representative samples of the population, e.g., [10], who also consider whether observer effects matter or not. However, that study primarily examines the stability of *individual characteristics* like risk-taking, confidence and cognitive abilities across populations,

with only a few interactive games (a dictator game and a prisoner's dilemma game). By contrast, in this study we compare whether there is any difference in the play of a *wider variety* of interactive laboratory games as well as some individual decision tasks all using mobile devices. In three of these games we vary the scale of the groups in which subjects interact with one another. We compare play of such games and tasks in person (as in a laboratory study where an observer is present) or remotely by the same sample of subjects, in our case university students and like [10] we find little evidence that observer effects matter.

The idea that group size can matter for experimental results has also been previously examined. In one of the earliest studies of the public goods game, [11] considered groups of size 4, 10, 40 and 100 and found that average contributions increased with the group size, e.g., from 10 to 100, but only if the marginal per capita return (MPCR) on contributions to the public good was low (0.30). [12] replicate this finding in comparisons of groups of size 4 and 8, but find that larger groups contribute less than smaller groups when the MPCR is high (0.75). [13] report on beauty contest games conducted among the readership of three newspapers resulting in sample sizes of 1,468 to 3,696 participants. [14] study voter turnout in the laboratory in small groups of size 3 as well as in larger groups of up to size 53, while [15] conduct a similar experiment involving groups of size 30 or 300 using an Mechanical Turk's online workforce. A main finding from all of these studies is that group size can matter for the behavior observed. Other experimental studies focusing on group size effects include [16, 17] who look at how large groups of players, up to size 100, play the volunteer's dilemma; [18–20] who look at asset pricing in markets with large number of subjects, between 40-300. Still, these studies are typically conducted on different dates in time or on different populations (laboratory subjects versus newspaper readers, combining several laboratories at once) or lack other elements of control, e.g., newspaper readers can discuss their choices with one another. Thus, an important contribution of our paper is that we conduct the large versus small group treatments simultaneously via random assignment of members of a single population to either the small or large group sizes. Thus, our design does not mix populations making better use of random assignment and therefore resulting in a more controlled test of group size effects. Nevertheless, we will compare our results on group size differences with the existing literature to examine whether there is any impact from collecting data on mobile platforms or whether data is gathered in-person versus remotely.

## Results

In this section, we present our main experimental results. In our analysis of the games, we follow the order in which the games were played in our experiment. Methodological details are provided in the Methods section and in the Section E in S1 File.

### Large versus small group size effects

The main treatment variable of our first 3 games, for which group size is scalable, is Large ($N = 100$) vs. Small ($N = 10$) group sizes. For each game or task, we relate our findings to those found in the literature.

**Beauty contest game.**  In our first game, initially studied by [21], participants in each group of size $N$ simultaneously and without communication guess a number in the interval [0, 100]. The winning guess is the number closest, in absolute value, to $p$ times the group average. We set $p = 2/3$ and the game was played for 3 repetitions (rounds) among unchanging members of a group of size 10 or 100. The winner in each group earned 20 points per round (the prize is evenly split for a tie); everyone else earned 0 points.

Regardless of group size, iterative elimination of dominated strategies results in the equilibrium prediction that all $N$ players will guess 0. We chose to include the beauty contest game in our experiment comparing large and small groups as this game is one for which the payoff incentives do not change as the size of the group gets larger.

Fig 1 shows the cumulative distributions of guesses in the 2/3-beauty contest game across experiments, treatments, and rounds. For comparison purposes we also include [21]'s lab data and [22]'s Financial Times experimental data for the 2/3 × the average treatment. We use Kolmogorov–Smirnov (KS) tests to show that guesses in our large group treatment are distributed lower and thus closer to the equilibrium prediction of 0 than in our small group treatment by round 2 of Experiment 1 ($KS = 0.2709$, $p < 0.001$) and by round 3 of both experiments (Experiment 1: $KS = 0.5125$, $p < 0.001$; Experiment 2: $KS = 0.2450$, $p < 0.001$), although the difference is not statistically significant in round 1 (Experiment 1: $KS = 0.1457$, $p = 0.080$; Experiment 2: $KS = 0.0924$, $p = 0.251$). We speculate that in larger groups, subjects may react to the greater competitive pressure by iterating their reasoning somewhat further than they would in smaller, less competitive groups. Alternatively, in larger groups the effects of outlier choices such as a guess of 100 may be more greatly diminished, and subjects may react to this difference by making guesses that are closer to the equilibrium prediction. In comparison with [21]'s data, we find using KS-tests that with the exception of round 1 of Experiment 1, there is no significant difference in the distribution of guesses over rounds 1-3 between [21]'s experimental data and the data from our Large group treatment (both Experiments 1 and 2). However, KS-tests reveal that the distributions of guesses in rounds 2-3 of our Small group treatment in both Experiments 1 and 2 are significantly different from [21]'s data in the direction of being further from the equilibrium prediction ($p < .05$ for all tests). Panel D of Fig 1 indicates that there are no differences in mean round 1 guesses between [21]'s data and any of our experiment/treatments, according to the bounds of 95% confidence intervals. Further, we do not find differences in mean guesses for rounds 2-3 in our large $N = 100$ treatment and [21]'s data with groups of size $15 - 18$. However, our small groups of size 10 have mean rounds 2-3 guesses that are greater than found in [21]'s data, significantly so in the case of Experiment 1. It may be that the 5-8 additional subjects in [21]'s experiment as compared with our small treatment group size of 10 are responsible for this difference.

Similarly, if we compare the distribution of guesses in our experiments and [21]'s data with that of [22]'s data from the one-shot *Financial Times* 2/3-the-average beauty contest game, where $N = 1,468$ (nearly 15 times our large group size) we find large and significant distributional differences. As panel A of Fig 1 clearly reveals, the distributions of guesses in all other experiments/treatments first order stochastically dominate the distribution of guesses found in Thaler's study which is a strong indicator that group size matters. What we add to the discussion is the demonstration that group size differences can become apparent as randomly formed groups of different sizes gain experience from repeated play, all using the same mobile platform to make choices and therefore reducing the role of possible confounding factors. Clearly, more research is needed on the precise threshold for which the size of a group begins to have statistically detectable effects on behavior in strategic interaction games such as the beauty contest.

Other studies (e.g. [13]) have found no statistically significant group size effects in this game, but they have typically compared different subject populations and/or one-shot games. The most similar study [23] compared guesses in a $p = .7$ beauty contest game by small ($N = 3$) versus large ($N = 7$) groups over 10 rounds and found that larger groups played the equilibrium prediction more often than did smaller groups, but their result was not statistically significant.

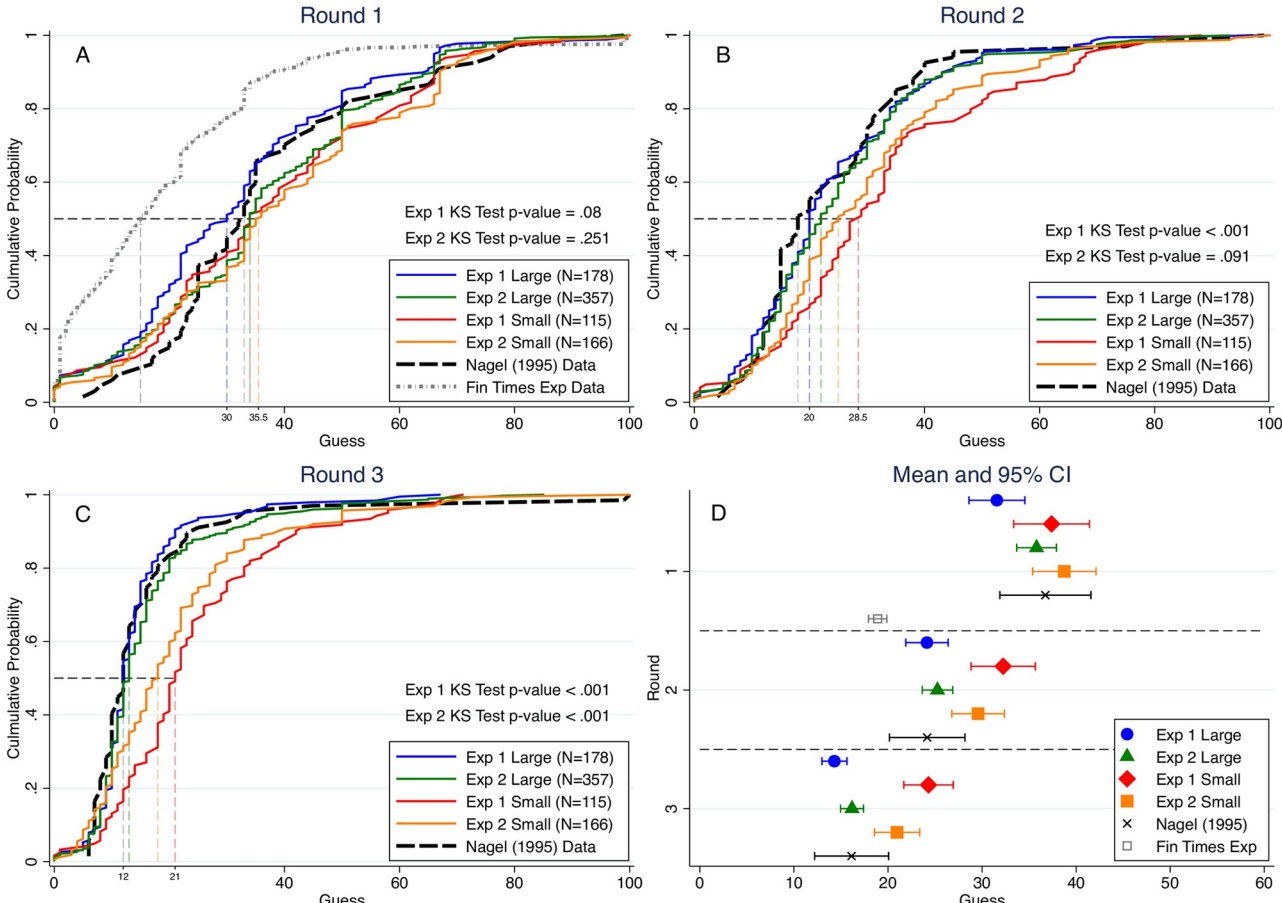

**Fig 1. Cumulative density function of guesses across different rounds.** (A-C) The distribution of guesses in large groups and small groups from Experiments 1 and 2 as well as from [21, 22]. The numbers of observations are shown in the legends and the median guesses of large and small groups are labeled in dashed lines. The *p*-values of the Kolmogorov–Smirnov tests for different groups are provided at the bottom of each figure. (D) The means and 95% CI for different groups from round 1 to round 3.

**Voter turnout game.** Our second game is based on the experimental voter turnout study of [14] in which there are two teams of different sizes, in a ratio of approximately 2:1 membership. In the Small groups treatment, the majority team has 7 members while the minority team has 3 members. In the Large groups treatment, the majority has 67 members to the minority's 33 members. Members of each team have to simultaneously and without communication decide whether or not to vote, which is costly. The team with the most votes wins a prize of 100 points, the losing team gets 0 points, and in the event of a tie, both teams get 50 points. Each individual's cost to voting is private information, known right before voting, and distributed uniformly over the interval [0, 80]. Thus, each individual's payoff was the team prize (100, 50, or 0) minus their voting cost if they voted or a cost of 0 if they abstained. The game was played three times by members of the same group, and subjects received feedback at the end of each round on the number of votes cast by both teams as well as the winning team and their own payoff for the round.

This voter turnout game has a number of testable predictions stemming from the Bayesian Nash equilibrium (hereafter, BNE). First, turnout for both teams should decline with increases in the total group size (from 10 to 100) as voters become less pivotal. Thus, turnout should be

smaller in the Large treatment as compared with the Small treatment. Second, regardless of group size, the turnout rate should be higher for the minority team as compared with the majority team in order to offset the size advantage of the latter group (this is also known as the "underdog effect").

Fig 2 shows mean turnout rates with 95% CIs and BNE predictions across experiments, treatments, and rounds. The last panel of this figure reports on data from two of [14]'s "landslide" treatments that are closest to our design, one with $N = 9$ and the other with $N = 51$ and both with a 2:1 ratio for membership in the two teams.

First, observed turnout is generally much greater than predicted for either team in both the Large and Small group treatments across both experiments. One exception is turnout by the minority team in the Small group treatment of Experiment 2, where, in rounds 2 and 3, we find no statistical difference between the mean and predicted turnout rates for the minority team (two-tailed $t$-test Round 2: $t(48) = 0.7996$, $p = 0.4279$; Round 3: $t(49) = 0.7093$, $p = 0.4815$).

Second, while in Experiment 1 turnout rates for the majority teams are significantly lower by round 3 in Large groups than in Small groups (Mann-Whitney test of differences, $p$-value = 0.0241), there are no corresponding differences for the minority teams across treatments. Similarly, in Experiment 2 we find no difference in turnout rates for majority or minority teams across treatments ($p > 0.100$ for all three rounds), except for minority team members in round 1. See Section B.2 in S1 File for a detailed analysis.

Third, counter to equilibrium predictions, we do not observe an underdog effect; turnout rates for the majority team are always higher in both the Large and Small group treatments than for the minority team. See S7 and S8 Tables in S1 File for details. This finding is nevertheless consistent with many other experimental team participation/voting game studies under majority rule e.g., [24–27]. As [27] points out (footnote 4), the only experimental paper reporting greater turnout by minority team members in a majority rule setting is [14].

Indeed, by comparison with the most comparable treatments from [14]'s experiment (LP), turnout is higher in both our Experiments 1 and 2 for both minority and majority team members with the exception of minority team members in the small group treatment of Experiment 2, rounds 2-3, and we do not observe the underdog effect. One explanation for this difference between our results and those of LP might be that in conducting multiple experiments at the same time, we are unable to give subjects as much instruction as in LP's study and we did not have two practice rounds or test subjects on their understanding of the instructions as LP did. Consequently, it may take longer for subjects to learn equilibrium play, but as noted above, we do seem some evidence for learning over time in our experimental data.

**Linear public goods game.**   Our third game is a linear public goods game based on [28]. In this game, subjects are assigned membership to a group of size $N = 10$ (small) or 100 (large). In each round, each group member, $i$, is endowed with $\omega$ tokens and must decide simultaneously and without communication how many tokens, $0 \leq x_i \leq \omega$ to contribute to a public good. Player $i$'s payoff in points is given by:

$$\pi_i = \omega - x_i + \beta \sum_{j=1}^{N} x_j \tag{1}$$

Our experiment used standard parameterizations from the literature, where $\omega = 20$ and the marginal per capita return (MPCR) $\beta = 0.3$. The game was played repeatedly for 8 or more rounds, but we truncate to 8 rounds for comparison across groups/experiments. Following each round of play, subjects received feedback on the group contribution and their own round payoff.

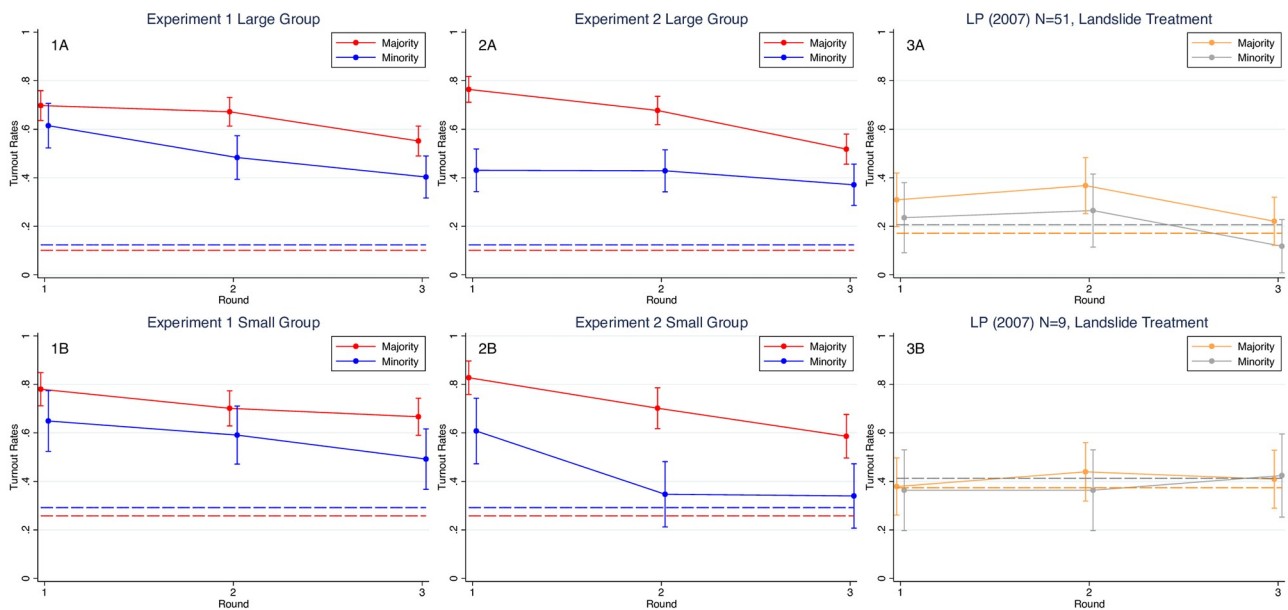

**Fig 2. Majority and minority team turnout rates under different electorate sizes.** (1A and 1B) The turnout rates for large and small groups from Experiment 1. (2A and 2B) The turnout rates for large and small groups from Experiment 2. (3A and 3B) The turnout rates in round 1 to round 3 for large (N = 51) and small (N = 9) treatments from [14]. The red (orange) and blue (gray) curves are the turnout rates for the majority and minority teams (with 95% CI), respectively. The dotted lines are the theoretical predictions of the majority and minority turnout rates.

The dominant strategy Nash equilibrium is that subjects contribute 0 to the public good in all rounds since the MPCR $\beta < 1$ on group contributions is less than the marginal return on investments made to the private account, which is equal to 1. However, it is <u>socially optimal</u> if all $N$ players contribute their entire endowment, since $N\beta > 1$. Provided that $N > 1/\beta$ (as was true in both our large and small group treatments) these predictions are invariant to the group size, though one might expect contributions to be lower in larger groups owing to the greater temptation to free ride on the contributions of others or the greater perceived social pressures in smaller groups to contribute more [29].

Fig 3 panels 1A-2A report on mean contributions to the public good as a percentage of subjects' endowment across experiments, treatments, and rounds. For comparison purposes these two figures also report on data from [11]'s (IWW) experimental treatments with large ($N = 100$) and small ($N = 10$) groups and an MPCR = 0.3.

As in all experimental tests of the public goods game, we find that contributions tend to decrease over time as subjects gain experience and observe the average group contributions. But in contrast to IWW, we find no difference in contributions between Large and Small groups over each of the 8 rounds as can be seen in the 95% CI bars for each round in Fig 3. We also do not find significant differences in average contributions between large and small groups over *all* rounds (Mann-Whitney tests with pooled data over all 8 rounds, Experiment 1: $p = 0.343$; Experiment 2: $p = 0.405$).

By contrast, IWW found significantly larger contributions in their large group experiment as opposed to their small group treatment (with $\beta = 0.3$) as Fig 3 reveals. Our data on contributions for both Large and Small groups are more in line with IWW's findings for contributions made by small groups alone. Indeed, with a few exceptions, we do not find any significant difference between contributions made by our Large or Small group participants and contributions made by IWW's Small group participants over the 8 rounds shown. Possible explanations

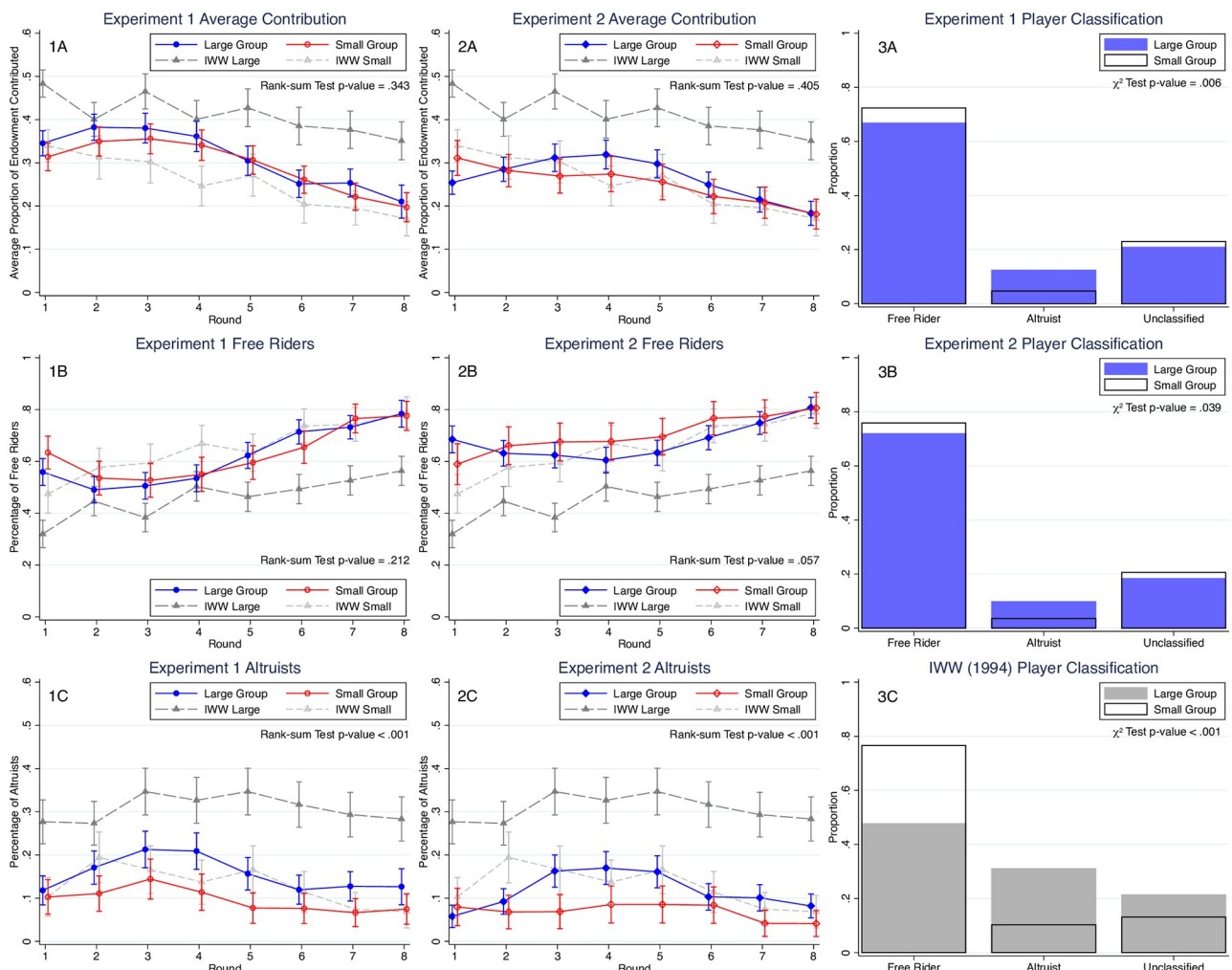

**Fig 3. Average contribution, proportion of free-riders and altruists and player classifications in the public good games.** (1A-1C) The average proportion of endowment contributed, proportion of free-riders and altruists (overlaid with 95% CI) across different rounds in Experiment 1 with the *p*-values from a Mann-Whitney test on pooled data over all eight rounds. (2A-2C) The average proportion of endowment contributed, proportion of free-riders and altruists (overlaid with 95% CI) across different rounds in Experiment 2 with the *p*-values from a Mann-Whitney test on pooled data over all eight rounds. Notice that in order to compare our results with the literature, we overlay the data from IWW in the 6 panels above. (3A-3C) The classification results of Experiments 1, 2, and IWW data with the *p*-values of $\chi^2$ tests.

for this result are that IWW did not provide monetary incentives to participants in the treatments reported on here. Instead, subjects received course extra credits. (Other differences are that IWW set $\omega = 25$ while we set $\omega = 20$, and IWW conducted rounds over several days while we do not). While IWW did not find significant differences between contributions made by players in small groups of 10 who were rewarded in extra credit points versus those who were incentivized (in earlier experiments reported on in [28]) using money payments, they do not report a similar comparison between the large group ($N = 100$) treatments they ran, where subjects were rewarded with extra credit points and treatments (not run) where subjects in the $N = 100$ group treatments earned monetary payments. Thus, it is possible that money payments, as in our design, reduce the amounts that participants are willing to contribute to the public good in the large group treatment, as this is where we find the greatest difference with IWW's findings.

More generally, our results on group size differences are generally consistent with the ambiguous effects of group size on public good contributions that is found in other papers in the literature. For instance, a meta-analysis of 27 linear public goods game experiments by [30] found insignificant effects of group size for public good contributions.

Fig 3 panels 3A-3C also reveal player classifications into two main behavioral types, free riders and altruists. Panels 1B-2B, and 1C-2C reveal differences in the patterns of contributions made by these different behavioral types over time between Large and Small groups. (See Section B.3 in S1 File for type definitions and a more detailed analysis). In Small groups we find a significantly larger proportion of free-riders while in Large groups we find a significantly larger proportion of altruists (Experiment 1: $\chi^2$ test $p$-value = .006; Experiment 2: $\chi^2$ test $p$-value = .039). We find similar patterns in our analysis of IWW's data where the differences in the proportions of the two types are statistically even greater ($\chi^2$ test $p$-value <.001), than in our data, which may further account for why we do not observe the large differences between large and small groups that they observe. These differences in the proportions of player types by group size are new and suggest that individual behavior may be quite malleable and dependent on the size of the group that subjects find themselves in. For instance, subjects in the large (small) group treatment might believe that free-riding is more likely (less likely) and might respond to such beliefs by contributing more (less). Future work on this topic would require a more careful, within-subject experimental design that varied the group size and elicited subjects' beliefs about the contributions of others.

Finally, we note that we did not adjust the scale of returns from public good contributions, with the change in the group size, that is, the total factor on public good contributions, $\beta \times N$ is 10 times greater in our large group treatment as compared with our small group treatment. To keep this factor constant while maintaining the dominant strategy of 0 contributions, we would have to lower the MPCR in our large group treatment from .30 to .03 (making two changes rather than the single change in group size). While other researchers have made such adjustments (see, e.g., [28]), this is less easy to do when subjects are participants in a single, synchronous session, where they are given a single set of instructions and are randomly assigned to small or large group settings as in our experiment. Despite the 10 fold increase in the payoff from full contribution in the large group treatment, we do not find significant increases in the fraction of subjects' endowments they are willing to contribute to the public good, which is surprising. Further, we do not find that free riding behavior is more pervasive in larger groups as compared with smaller groups and in fact, the evidence seems to point to the opposite conclusion.

## Other games and tasks

In both experiments, the first three games were followed by five 2-player or individual decision-making games/tasks. The order of games/tasks 4-8 was as follows: 4) a 2-player ultimatum game played once (in Experiment 2 we reversed the roles and played the game a second time), 5) an individual Holt-Laury type risk preference elicitation played once, 6) a 2-player centipede game played 3 times, 7) a 2-player trust game played once (in Experiment 2 we reversed the roles and played it a second time), 8) an individual 3-round real effort task exploring gender differences in compensation schemes. Details and results for these games are in Section B in S1 File. Here we briefly highlight how the five games in our experiments replicate previous findings from lab experiments.

**Ultimatum game.** Our 2-player ultimatum game follows [1]. We find that offers from the proposer to the responder are mostly 50 percent of the pie or less, and the conditional acceptance rate is monotonically increasing in the proposed offer amount, which is consistent with

[31, 32]. A direct comparison of our dataset and that from [31] generates similar results in the distribution of proposal offers and conditional acceptance rate (see Section B in S1 File for details). Moreover, we find that in Experiment 1, the relative reaction time for acceptance peaks near the offer size at which subjects are equal likely to accept or reject (around 30%), and the reaction time falls when accepting higher offers or rejecting lower offers. This inverted U-shape relationship between the proposal offer and the reaction time is consistent with [33–35].

**Risk elicitation.** Our risk elicitation task was based on a multiple price list similar to [2]. We find risk averse players choose 1.395 and 1.891 more safe options than risk neutral players in Experiments 1 and 2, respectively, which is consistent with [2]. But in Experiment 2 we find that players are significantly more risk averse than players in Experiment 1 ($KS = 0.1341$, $p < 0.001$). Only a small percent of risk-seeking players were observed (11.6% in Experiment 1 and 13.0% in Experiment 2), which is also consistent with [2] among other studies.

**Centipede game.** Our four-node centipede game follows [3]. The subgame perfect Nash equilibrium (SPNE) is that the first mover chooses Take at the first opportunity ending the game. However, [3] report that only 7% of the first movers played the SPNE. The most common experimental finding is that the frequency of "Take" increases as inexperienced subjects move closer to the terminal decision node, while experienced subjects who have played at least once, learn to choose Take earlier and earlier with repetition.

Our finding is consistent with the literature. The frequency of Take ("the take rate") is significantly greater at decision node 2 than at node 1 in the first two rounds. By the third round, however, the take rates at nodes 1 and 2 are the same at 50%, similar to [3]. We note that the take rates of 28.3% and 24.6% at node 1 in our two experiments are higher than the 7.1% rate of [3].

**Trust game.** The trust game in our playlist was originally studied by [4]. The SPNE of this one-shot game (as in our study) is for the responder to always return 0 and thus for the investors to always send 0. However, [4] found that on average investors sent 51.6% of their endowment, while responders returned about 90% of the amount invested.

We observe similar, but slightly different results in our experiment. The mean investments out of an endowment of 100 were 29.02 and 38.48 in the two respective experiments, the distributions of which are significantly different ($KS = 0.1669$, $p < 0.001$). The mean ratios of the return to the investment in the two experiments are 60.24% and 46.53%, both of which are less than the 90% in [4], which may be attributed to the fact that the responder has no endowment in our setup. We also find the return rate decreases with the investment. A no-intercept regression of the return on the investment has a significantly positive slope of 0.707 ($p < 0.001$) for Experiment 1 and 0.680 ($p < 0.001$) for Experiment 2.

**Math competition.** Following [5], we implemented our last game called "Math Competition" to explore gender differences in the choice between piece-rate and tournament compensation schemes. The main finding from [5] is that while there is no difference in the performance between men and women in the math task in any given compensation setting, when offered the choice of compensation schemes, women disproportionately prefer the piece rate scheme while men prefer the tournament compensation structure. [5] report that 73% of men choose the tournament but only 35% of women make that choice.

While different in several respects from [5], our experiments generate qualitatively similar results. In Experiment 1, 62.2% of men choose the tournament while only 46.4% of women do the same (Mann-Whitney Rank-sum Test: $p < 0.001$). Further, conditioned on *losing* the tournament in the first stage of the no-choice setting, men are significantly more likely to choose the tournament than are women ($p < 0.001$). However, we find no gender difference ($p > 0.10$) conditioned on *winning* the tournament previously. We have a set of similar results in Experiment 2.

## Regression analysis

In this section we report on a regression analysis examining the relationship between subjects' choices and potential individual explanatory factors including each subject's Gaokao score, gender, CRT score, and elicited risk aversion, among other variables that we collected as part of both experiments. For the one-shot games, we simply adopt the OLS model to analyze correlations. For the other games that were played repeatedly, we further use a convergence model proposed by [36] to estimate long-run behavior, reported at the bottom of Table 1. This time series model is useful for understanding dynamic behavior as it explores the role of time (or experience). Specifically, for the outcome variable of interest in Experiment $i \in \{1,2\}$ this model is:

$$y_{it} = \alpha + (1/t)\mathbb{X}_i \cdot \beta_1 + (1 - 1/t)(\beta_2 \text{Large}_i + \beta_3 \mathbb{1}\{i = 2\}) + \epsilon_{it}, \tag{2}$$

where $y_{it}$ is the outcome variable in period $t$, $\mathbb{X}_i$ collects the demographic variables of the subjects in Experiment $i$ and $Large_i$ is a dummy variable that takes the value of 1 for large groups and 0 otherwise. Therefore, $\beta_1$ reveals the initial position of a possible convergence process, since initially, when $t = 1$ the value of the dependent variable in Experiment $i$ is equal to $\alpha + \mathbb{X}_i \cdot \beta_1$. Similarly, $\beta_2$ and $\beta_3$ show the asymptotes of the dependent variable for large groups and Experiment 2, respectively; as $t$ gets larger the weight on $\beta_1$ diminishes while the weight on $\beta_2$ and $\beta_3$ gets larger since $(t - 1)/t$ approaches 1.

For the beauty contest game, the first column of Table 1 reveals that in Experiment 1, small groups appear to be converging to a guess of 19.319, while large groups appear to be converging to a guess of 13.675. Further, the asymptotes for Experiment 2 would be adjusted downward by 5.224. For the voter turnout game (the second column of the table), although players in large groups are initially less likely to follow the BNE, the group size effect vanishes in the long run while players in Experiment 2 are 12.09% more likely to follow BNE. In the public goods game (columns 3-5) players in large groups are asymptotically more likely to be strong free riders by 7.56% and altruists by 8.72%. However, players in large groups are 5.09% less likely to be a free rider suggesting that large groups have more extreme players in the long run. In the centipede game (columns 6-7) there is no difference in group sizes and for the two experiments in the estimated asymptotes, though a higher CRT score significantly reduces the probability that a subject chooses Take in the second node.

Table 2 summarizes the regressions for the one-shot games. For the trust game (regression specifications 1-2), female subjects invest less on average and subjects in Experiment 2 invest more. In addition, specification (2) shows that players with higher CRT scores tend to return more. For the Ultimatum game (regression specifications 3-4), female proposers offer marginally more than do males. Furthermore, more risk averse proposers are more likely to make a higher proposal. From a logistic regression, we find that the responder's acceptance rate is monotonically increasing in the proposal offer and Experiment 2 has a higher conditional acceptance rate. Regarding the choice of tournament or piece rate in the individual choice experiment where subjects first have experience with both compensation schemes, we confirm that females are significantly less likely to choose the tournament than males and we find that risk averse subjects are also less likely to choose the tournament. On the other hand, winning the tournament (in Round 2) makes it more likely that a subject will choose the tournament scheme, particularly for males.

## Discussion

We have demonstrated the possibility of conducting large, repeated, multi-game economic experiments with subjects' own mobile devices and making use of mobile payments. Our

**Table 1. Estimation results of the convergence model.**

| Dependent Variable | (1) | (2) | (3) | (4) | (5) | (6) | (7) |
|---|---|---|---|---|---|---|---|
| | Guess | Pr(FollowBNE) | Pr(StrongFR) | Pr(FR) | Pr(Altruist) | Pr(Takeat1) | Pr(Takeat2\|Passat1) |
| Constant | 19.319*** | 0.573*** | 0.229*** | 0.685*** | 0.089*** | 0.561*** | 0.608*** |
| | (1.816) | (0.044) | (0.017) | (0.019) | (0.011) | (0.058) | (0.092) |
| *Initial Point* | | | | | | | |
| Initial Constant | 23.838*** | -0.168 | -0.012 | -0.380** | 0.182** | -0.252 | -0.143 |
| | (5.724) | (0.124) | (0.089) | (0.127) | (0.080) | (0.156) | (0.220) |
| Large Group | -1.529 | -0.077** | -0.039 | -0.021 | 0.007 | 0.038 | -0.004 |
| | (1.636) | (0.036) | (0.026) | (0.036) | (0.021) | (0.044) | (0.061) |
| Female | 5.631*** | -0.086** | -0.295*** | -0.104*** | -0.078*** | 0.013 | 0.014 |
| | (1.568) | (0.037) | (0.032) | (0.037) | (0.027) | (0.047) | (0.067) |
| Gaokao Score | -4.765*** | -0.008 | -0.051* | -0.071** | 0.053* | 0.052 | 0.034 |
| | (1.317) | (0.033) | (0.027) | (0.032) | (0.031) | (0.048) | (0.061) |
| CRT | -4.898 | 0.153** | 0.068 | 0.211*** | 0.031 | 0.089 | -0.346** |
| | (3.479) | (0.071) | (0.057) | (0.081) | (0.048) | (0.091) | (0.135) |
| Risk Aversion | -0.257 | 0.002 | -0.031*** | -0.012* | -0.003 | 0.004 | 0.022* |
| | (0.294) | (0.007) | (0.006) | (0.007) | (0.005) | (0.009) | (0.011) |
| Experiment 2 | -0.804 | -0.047 | 0.115*** | 0.111*** | -0.058*** | -0.067 | -0.107* |
| | (1.678) | (0.035) | (0.026) | (0.035) | (0.022) | (0.043) | (0.062) |
| School Province FE | Yes | Yes | Yes | Yes | Yes | Yes | Yes |
| Birthplace Province FE | Yes | Yes | Yes | Yes | Yes | Yes | Yes |
| *Ending Point* | | | | | | | |
| Asymptote × Large Group | -5.644*** | -0.006 | 0.076*** | -0.051** | 0.087*** | 0.011 | -0.074 |
| | (1.745) | (0.048) | (0.019) | (0.021) | (0.013) | (0.065) | (0.095) |
| Asymptote × Experiment 2 | -5.224 | 0.121*** | 0.048** | 0.079*** | -0.039*** | 0.043 | -0.012 |
| | (1.700) | (0.046) | (0.019) | (0.021) | (0.014) | (0.063) | (0.092) |
| Observation | 2,259 | 2,735 | 7,320 | 7,320 | 7,320 | 1,487 | 800 |
| R-squared | 0.2266 | 0.0529 | 0.0694 | 0.0438 | 0.0304 | 0.0741 | 0.1047 |

Note: Standard errors are corrected for AR(1) and are shown in parentheses.

* $p < 0.10$;

** $p < 0.05$;

*** $p < 0.01$.

results, particularly for small groups and 2-player games or individual decision making tasks are similar to findings from traditional laboratory experiments, which provides reassurance that our mobile platform does not appear to affect subject behavior. We have further demonstrated that our approach does not require that subjects participate in-person; we have repeated our experiment a second time with subjects playing the same set of games remotely, and we obtain similar results. This replication of our original findings one year later with similar results builds confidence in the robustness of our findings and suggests that differences in the degree of physical control and/or social context between participating in-person or remotely may be less impactful on subject behavior for the scenarios we tested than was previously thought.

At the same time, we are able to leverage our large-scale, multi-game approach to obtain interesting *new* findings on group size effects and how individual characteristics may play a role in behavior across games that would be difficult to obtain in traditional, limited capacity laboratory settings. Our methodology provides an exciting and promising way forward for

**Table 2. Regression table for one-shot games.**

| Dependent Variable | (1) | (2) | (3) | (4) | (5) |
|---|---|---|---|---|---|
| | Investment | Return | Offer | Acceptance | *Pr(Tournament)* |
| Constant | 43.791*** | | 36.705*** | -6.067*** | 0.085 |
| | (13.311) | | (5.329) | (1.286) | (0.664) |
| Large Group | 3.852 | | 0.320 | 0.247 | -0.118 |
| | (3.270) | | (1.502) | (0.387) | (0.180) |
| Female | -14.855*** | | 3.513* | -0.043 | -0.563** |
| | (4.347) | | (1.941) | (0.415) | (0.241) |
| Gaokao Score | -0.499 | | -1.892 | 0.583 | -0.334* |
| | (3.522) | | (1.888) | (0.405) | (0.171) |
| CRT | 7.736 | | -6.572* | 1.213 | -0.414 |
| | (8.166) | | (3.878) | (0.844) | (0.436) |
| Risk Aversion | -1.248* | | 0.710** | 0.014 | -0.131*** |
| | (0.721) | | (0.358) | (0.073) | (0.037) |
| Experiment 2 | 8.760** | | 0.262 | 1.329*** | 0.366** |
| | (3.628) | | (1.658) | (0.393) | (0.173) |
| Investment | | -0.208 | | | |
| | | (0.267) | | | |
| Investment × Large Group | | -0.197 | | | |
| | | (0.121) | | | |
| Investment × Female | | -0.250 | | | |
| | | (0.173) | | | |
| Investment × Gaokao Score | | -0.001 | | | |
| | | (0.097) | | | |
| Investment × CRT | | 0.705*** | | | |
| | | (0.264) | | | |
| Investment × Risk Aversion | | 0.000 | | | |
| | | (0.028) | | | |
| Investment × Experiment 2 | | -0.064 | | | |
| | | (0.115) | | | |
| Offer | | | | 0.153*** | |
| | | | | (0.017) | |
| Gameplay Math Score | | | | | 0.328*** |
| | | | | | (0.112) |
| Win in Round 2 | | | | | 1.892*** |
| | | | | | (0.365) |
| Win in Round 2 × Female | | | | | 0.772* |
| | | | | | (0.406) |
| School Province FE | Yes | Yes | Yes | Yes | Yes |
| Birthplace Province FE | Yes | Yes | Yes | Yes | Yes |
| Observation | 493 | 453 | 489 | 450 | 964 |
| R-squared | 0.1479 | 0.6959 | 0.1658 | 0.4881 | 0.2728 |

Note: We apply the logistic regression model to models (4) and (5). The reported $R^2$ is pseudo $R^2$. The robust standard errors are shown in parentheses.

* $p < 0.10$;

** $p < 0.05$;

*** $p < 0.01$.

experimental research, not only for classic lab-type experimentation but also for randomized control trial (RCT) experiments conducted in the field and remotely online. Indeed, our approach blurs the difference between laboratory and field RCTs.

Experiments hosted on the cloud can scale efficiently to run with hundreds or thousands of in-person or remote participants. Whether such scaling also require an adjustment to payoffs is an important consideration that will depend on the game being studied as we have noted. Nevertheless, user interfaces and incentives can be kept consistent across device types to ensure comparability of results and minimization of other confounding factors, such as the manner in which data are collected. Combined with mobile payments, our tools and approach can significantly lower the costs of and create new opportunities for running laboratory, field, and online experiments.

We recognize that there are rival, labor-saving ways of implementing the large scale, repeated multi-game experiment that we conducted and report on in this paper. For instance, social science researchers can use and have used on-line workers (e.g., on Amazon's mechanical Turk) or large, on-line panels of subjects (e.g., Prime panels). However, using common-pool online workforces also has its drawbacks; worker payments are low and set by market-wide conditions rather than by the experimenter; there is the risk of bot players, or players playing on multiple accounts at the same time leading to more screening of subjects and cleaning of data; experimenters typically have less control over participants' attention, participation, and interactions; finally, players on such platforms might be very experienced (less naive, less pro-social) about social science research questions than is the general population. Our approach allows for the quick recruitment of any sample of subjects (including the traditional sample of university student subjects), with fewer of the downsides of online workforces or panels.

Our findings provide the first important proof-of-concept for a new methodology of controlled experimentation using mobile platforms and payments. We hope that others build upon our approach.

## Methods

### Participants

For both Experiments 1 and 2, subjects were recruited from a group of college students participating in an economic summer camp at Xiamen University. Participants were students who just completed their third year of university study and wanted to go to graduate school for a master or Ph.D degree. They applied with references from their own university and were selected by the School of Economics at Xiamen University for a 5-day summer camp. The summer camp includes the first 2-day lectures by faculty members to introduce all the fields in economics at Xiamen University and the following 3-day exams for the qualification of entering the graduate programs of economics at Xiamen University without taking the national entry exams for graduate schools. The summer in 2019 was held in person. Yet, due to COVID-19, the summer camp in 2020 and Experiment 2 were held online.

There were 633 players in Experiment 1 and 585 players in Experiment 2 who participated in at least one of the eight games. At the beginning of both experiments, we communicated to subjects that their decisions and corresponding points earned from all games would matter for their final payment. They were told to expect a show-up payment of 10 CNY for participating in the experiment and an additional payment based on the outcome of the experiment. Participants were told before the experiments that their participation in the experiment was completely voluntary and their decision to participate or not would not affect the evaluation of their performance in the summer camp. Our experiment did not cause participants any

adverse physiological or psychological reactions; it only collects data on simple economic decision-making problems. So the internal review board waived the need for written informed consent from the participants. The ethical approval of our experiment was obtained from the Internal Review Board of Experimental Teaching and from the Organizing Committee of the Summer Camp of the School of Economics, and the Managing Department (Division) of Social Sciences at Xiamen University.

## Implementation

Our experiments were conducted using MobLab, an online, cloud-based educational platform for conducting economics experiments using web browsers and/or mobile devices. We initially had only about 20 days advance notice to design and implement an experiment that could make use of the population of more than 600 students attending the 2019 summer camp. The eight games used in the experiment were pre-programmed by MobLab and selected for their ability to scale well to large groups as well as their potential to offer interesting cross-game correlations in behavior and outcomes to study. The design and implementation of the experiment was reviewed and approved by the organizing committee of the summer camp from the School of Economics, and the Managing Department (Division) of Social Sciences at Xiamen University.

In both Experiment 1 and in the remotely conducted Experiment 2, we presented game-specific instruction slides before the start of each game so that subjects had no prior knowledge of subsequent games. The screenshots, instructions, and configurations of the games can be found in Section E in S1 File.

On the day before the experiment, a survey was sent to the participants to collect demographic data on gender, place of origin, cognitive reflection test (CRT) scores and participant's (self-reported) score on China's National College Entrance Examination (NCEE), commonly known as the Gaokao score.

Experiment 1 took place in an auditorium with a capacity of 800 seats. Before the experiment, students were randomly assigned to two matching pools, section A (large groups) and section B (small groups), and seated in separate areas of the auditorium. It was made common knowledge that the players would only be matched with other players within the same section (either A or B) through all eight games, and group membership was shuffled between but not within games.

The 2020 Summer Camp and Experiment 2 were held online due to the COVID-19 pandemic. The recruitment procedure was the same as in Experiment 1 while the implementation of the randomization was different. Participants were randomly assigned and invited by email to join two separate MobLab classes. Players would only match against other players within their class / matching group. As in Experiment 1, it is common knowledge to all subjects that they would only be matched with the subjects in the same matching group while being re-matched before each game. See Section E.1 in S1 File for details of the implementation.

The average payoff per game was 3 CNY with a final payment being the sum of participant's show-up payment and total payoff across all 8 games. In Experiments 1 and 2, the overall average total payment was 37.61 CNY ($\approx$ 5.42 USD) and 40.00 CNY ($\approx$ 5.77 USD), respectively, or roughly the equivalent of 2 hours of work as an undergrad TA in China. Subjects were paid on Alipay, the payments platform of Alibaba, which is ubiquitous in China and is also the world's largest mobile payment platform. Their account information was collected before the experiment with consent and was only used for this experiment. Due to technical issues and frustrations faced by participants during some parts of Experiment 1, we decided to increase and smooth the average payment per participant, ex post. For each successfully completed game, a

participant was awarded a payoff of 0.0205 CNY × the points earned in that game. For games that could not be joined or completed, they were awarded the average payoff earned by participants who had successfully completed that game. The final participant payoff was the the sum of their show-up payment of 10 CNY and their game based payment with a minimum payment floor of 27 CNY for all participants.

A weakness of our within-subjects design is that the order of play of the games may matter, for instance there may be spillover effects from one game to the next. However, our ability to replicate many results under somewhat weaker control conditions than is typically employed, speaks to the robustness of those experimental findings.

## Supporting information

**S1 File. This document contains all of the supporting materials for this article.** Supporting materials contain S1–S22 Tables, S1–S27 Figs, and the experimental instructions. (PDF)

## Acknowledgments

We thank Colin Camerer, Joseph Tao-yi Wang, Thomas Palfrey, and audiences at the University of Queensland Behavioural and Economics Science Cluster e-seminar for helpful feedback. We also thank the MobLab team for developing and supplying the experimental platform and providing technical assistance both on-site and remotely throughout our experiments. Finally, we thank the student assistants, the administrative and technical staff, and the organizing committee of the summer camp from Xiamen University for helping with the preparation, organization, testing, and logistics.

## Author Contributions

**Conceptualization:** Zhi Li, Po-Hsuan Lin, Si-Yuan Kong, Dongwu Wang, John Duffy.

**Formal analysis:** Zhi Li, Po-Hsuan Lin, Si-Yuan Kong, Dongwu Wang, John Duffy.

**Funding acquisition:** Zhi Li.

**Investigation:** Zhi Li, Po-Hsuan Lin, Si-Yuan Kong, Dongwu Wang, John Duffy.

**Methodology:** Zhi Li, Po-Hsuan Lin, Si-Yuan Kong, Dongwu Wang, John Duffy.

**Project administration:** Zhi Li.

**Software:** Si-Yuan Kong, Dongwu Wang.

**Visualization:** Po-Hsuan Lin.

**Writing – original draft:** Zhi Li, Po-Hsuan Lin, Si-Yuan Kong, Dongwu Wang, John Duffy.

**Writing – review & editing:** Zhi Li, Po-Hsuan Lin, Si-Yuan Kong, Dongwu Wang, John Duffy.

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
