## [Decision Letter · Decision Letter 0]

5 Feb 2021

PONE-D-20-38803

Conducting Large, Repeated Multi-Game Economic Experiments Using Mobile Platforms

PLOS ONE

Dear John,

Thank you for submitting your manuscript to PLOS ONE. I've managed to collect opinions of your manuscript from three very well qualified readers. They all like the manuscript, but also all suggest that the paper could be made considerably better with attention to several issues. Prominent concerns include a call for some clarification of the paper's specific methodological contribution, a more careful comparison of results from your large sample remote sessions with smaller scale laboratory experiments, and more discussion of some of the potential costs of increasing the number of participants (such as payoff scalability). The readers also all seem to take issue with your section comparing performance across games.

 At base, the readers  all regard large scale remote experiments as an important possible direction for the future of experimental economics. The common thread of their comments is that they would like  a more thorough consideration of what this alternative approach may offer, when it may be useful and what costs it may impose.  In light of their comments, I invite you to submit a revised version of the manuscript that addresses their concerns.  

We look forward to receiving your revised manuscript.

Kind regards,

Douglas D. Davis

Academic Editor

PLOS ONE

Journal Requirements:

"Funding: Z.L. was supported by National Natural Science Foundation of China (Grant No. 71873116) and the NSFC Basic Science Center Program (Grant No. 71988101). Experiment 1 was funded by MobLab. Experiment 2 was funded by Xiamen University."

We note that one or more of the authors have an affiliation to the commercial funders of this research study : MobLab Inc.

3.1. Please provide an amended Funding Statement declaring this commercial affiliation, as well as a statement regarding the Role of Funders in your study. If the funding organization did not play a role in the study design, data collection and analysis, decision to publish, or preparation of the manuscript and only provided financial support in the form of authors' salaries and/or research materials, please review your statements relating to the author contributions, and ensure you have specifically and accurately indicated the role(s) that these authors had in your study. You can update author roles in the Author Contributions section of the online submission form.

3.2. Please also provide an updated Competing Interests Statement declaring this commercial affiliation along with any other relevant declarations relating to employment, consultancy, patents, products in development, or marketed products, etc.  

6. We note that Figure S1 and S2 in your submission contain map images which may be copyrighted. All PLOS content is published under the Creative Commons Attribution License (CC BY 4.0), which means that the manuscript, images, and Supporting Information files will be freely available online, and any third party is permitted to access, download, copy, distribute, and use these materials in any way, even commercially, with proper attribution. For these reasons, we cannot publish previously copyrighted maps or satellite images created using proprietary data, such as Google software (Google Maps, Street View, and Earth). For more information, see our copyright guidelines: http://journals.plos.org/plosone/s/licenses-and-copyright.

6.1.    You may seek permission from the original copyright holder of Figure S1 and S2 to publish the content specifically under the CC BY 4.0 license. 

6.2.    If you are unable to obtain permission from the original copyright holder to publish these figures under the CC BY 4.0 license or if the copyright holder’s requirements are incompatible with the CC BY 4.0 license, please either i) remove the figure or ii) supply a replacement figure that complies with the CC BY 4.0 license. Please check copyright information on all replacement figures and update the figure caption with source information. If applicable, please specify in the figure caption text when a figure is similar but not identical to the original image and is therefore for illustrative purposes only.

<h1>** **</h1>

Reviewers' comments:

Reviewer's Responses to Questions

**Comments to the Author**

1. Is the manuscript technically sound, and do the data support the conclusions?

Reviewer #1: Partly

Reviewer #2: Yes

Reviewer #3: Partly

2. Has the statistical analysis been performed appropriately and rigorously? 

Reviewer #1: Yes

Reviewer #2: Yes

Reviewer #3: Yes

3. Have the authors made all data underlying the findings in their manuscript fully available?

Reviewer #1: Yes

Reviewer #2: Yes

Reviewer #3: Yes

4. Is the manuscript presented in an intelligible fashion and written in standard English?

Reviewer #1: Yes

Reviewer #2: Yes

Reviewer #3: Yes

5. Review Comments to the Author

Reviewer #1: I selected "partly" for question 1, because the claim that the paper is a breakthrough in the way we conduct experiments is not supported. Please see the attached pdf document, in which I explain that the focus of the paper should instead be on the novel results.

Reviewer #2: I think this is an interesting paper. The paper claims to make the following contributions: 1) providing a proof-of-concept that large economics experiments can be conducted on mobile platforms, 2) exploiting the large participant pool to study the effect of large group sizes on behavior in games, 3) studying the correlation between choices and behaviors across different tasks. While the third contribution feels a bit disconnected from the others, I think the first two contributions are potentially very important. I think it is reasonable to assume that we will see a continuous shift away from the traditional lab and towards mobile platforms in experimental economics. Against this backdrop, this paper can provide useful evidence on the robustness of experimental methods and results to new mobile-based protocols. It also suggests ways in which the new technologies can be fruitfully employed for research in experimental economics, namely leveraging the large number of participants available in online studies.

My main concern is that to actually provide a useful reference for future mobile-based studies, this paper should provide a clearer comparison with standard lab experiments. The paper repeatedly argues that the experiments replicate several existing findings for small groups (see the abstract for example), but I do not think the paper actually shows this. For each of the three games, the paper comments on whether the effect of group size is consistent with estimates previously obtained in other studies. However, the paper does not discuss whether the results from the small groups treatment are similar or different from those of traditional lab studies. So, I would like the paper to provide a better analysis of whether the experiments replicate existing findings for small groups, for each of the three games and the other tasks as well. The comparison with previous lab studies should be done separately for experiment 1 and experiment 2. This can give the readers an idea of how the results are affected by whether the experiment is run in the traditional lab setting vs. in-person on mobile platforms vs. remotely on mobile platforms. I do not require a fully quantitative comparison (i.e. formal hypothesis testing). Below I describe exactly which points should be addressed for each game in the experiment.

1. Beauty contest game

a. How does the distribution of guesses observed in the small-group experiment compare to that in previous lab experiments with a similar parametrization?

b. How do the changes in average guesses across rounds compare to the dynamics of average guesses in previous lab experiments?

2. Voter turnout game

a. The paper finds that turnout is generally much greater than predicted by theory. Is this consistent with previous findings from lab experiments?

b. How do turnout rates observed in the small-group experiment (across the majority and minority teams) compare to those in previous lab experiments with a similar parametrization?

3. Public good game

a. How does the average contribution observed in the small-group experiment compare to that in previous lab experiments with a similar parametrization?

b. How does the distribution of player types in the small-group experiment compare to that in previous lab experiments?

c. How do the changes in the average contribution across rounds compare to the dynamics of average contributions in previous lab experiments?

4. Other games and tasks

I think it would be useful to highlight whether the results of the other games in the experiments replicate previous findings from lab experiments. Currently the results of the other games are presented in the supplementary materials. I have also noticed that, for most of these games, the supplementary materials already include a comparison with previous findings. I think these comparisons should be included in the main body of the paper. Note that I am not recommending to move the complete analysis of results in the main body, but I would like the paper to briefly highlight to what extend the remaining games in experiment 1 and experiment 2 replicate previous findings.

5. If after making the revisions suggested above the length of the paper exceeds the limit, I recommend shortening the section on “Behavior across games.” While this section is interesting, studying correlations across games feels a bit disconnected from the main contributions of the paper (which are comparing mobile-based experiments to traditional lab experiments and exploiting the large subject pool of mobile-based experiments).

Reviewer #3: Review of Conducting large, repeated, multi-game economic experiments using mobile platforms by Li et al. 2021.

The authors conduct a series of large incentivized experiments using a mobile platform (MobLab) in China. The experiments show that such large experiments are feasible. Some economic intuitions build on the intuition that single agents are unlikely to be consequential. The current research suggests some of these intuitions might be tested using online platforms.

Comments:

1. The paper is a nice illustration of what the future of experimental economics might look like. Several authors (e.g. Snowberg, Taubinsky, etc.) are embedding incentivized experiments in representative surveys. The paper shows that it might be possible to move from individual decision making to interactive decision making. This is a qualitative leap. The authors illustrate that there might be some solutions around the corner. Of course, this should not be surprising for any person vaguely familiar with the gaming industry. The contribution of the paper is to show that large experiments can be conducted with samples familiar to experimentalist (college students). This is important because it allows to compare results from standard lab sessions with virtual sessions (albeit in the experiments reported in the paper, subjects seem to be `captured’ subjects). The paper should emphasize more the fact that the only thing they are varying is session size.

2. An important concern with lab experiments is the nuisance costs associated with registering for an experiment, participating in an experiment, getting paid, etc. There is always a concern that those showing-up in an experiment are different from the population at large. The recent work by Snowberg and Yaariv (Testing the Waters: Behavior across Participant Pools) reiterates the fact that behavior is similar across subject pools. This paper adds to that evidence, but it provides a different kind of robustness test. A discussion of these alternative robustness test would be useful.

3. I do not have major comments on the studies conducted given they were meant to be proof-of-concept. However, the results could be presented on the light of recent experimental evidence. For instance, the behavior in the public goods game deserves more attention since the potential gains from cooperation in the large sessions is much more salient. The payoff function used in the paper would produce payoffs that are 10 times larger if subjects were to coordinate on the Pareto efficient outcome. Since the chosen strategies are similar across small and large sessions, subjects in large sessions would have experienced much larger payoffs. In a sense, the failure to coordinate is much more consequential.

4. The paper should discuss the scaling of payoffs more in detail. For instance, in the beauty contest, the scale of payoffs can be held constant regardless of the size of the group. So, one can evaluate the effect of group size directly. However, in public good games, the scale of payoff changes with the group size. So, changing the group size keeping constant the parameters of the payoff function do change the scale of payoffs. In other words, one would like to evaluate the effect of changing the parameters of the payoff function holding constant different group sizes (as in Isaac & Walker, 1988). This kind of methodological concern becomes more prominent as the opportunity to scale up the size of experiment emerges. The authors have an opportunity to take a fresh look at these issues in light of their results.

5. I find the section discussing behavior across games distracting. First, analysis across games using very large samples is provided by Rubinstein (QJE 2016) where a typology of players is attempted. Empirical relations are very interesting, but it is not clear what is really new in this section. A more detailed discussion of behavior in the large-scale games would be more informative since those are new and puzzling. In this case, there are clear theoretical benchmarks.

6. PLOS authors have the option to publish the peer review history of their article (what does this mean?). If published, this will include your full peer review and any attached files.

Reviewer #1: No

Reviewer #2: No

Reviewer #3: No

---

## [Author Response · Author response to Decision Letter 0]

24 Mar 2021

Our response to the editor and reviewers was uploaded as a pdf file, labeled "plos_one_response_letter.pdf"

---

## [Decision Letter · Decision Letter 1]

12 Apr 2021

Conducting Large, Repeated Multi-Game Economic Experiments Using Mobile Platforms

PONE-D-20-38803R1

Hi John,  

I've heard back from two of the three referees who read the initial version of your manuscript, and they agree that you have satisfactorily addressed the concerns they had regarding the initial version.  For that reason I am pleased to provisionally accept your manuscript for publication.  One referee suggests that you may be overstating your case a bit by classifying your results as a 'methodological breakthrough', nevertheless this reader does not insist on further changes and I will leave it to you to decide whether or not to make any further adjustments.

This notice indicates that your manuscript has been judged scientifically suitable for publication and will be formally accepted for publication once it meets all outstanding technical requirements.

Kind regards,

Douglas D. Davis

Academic Editor

PLOS ONE

Additional Editor Comments (optional):

Reviewers' comments:

Reviewer's Responses to Questions

**Comments to the Author**

1. If the authors have adequately addressed your comments raised in a previous round of review and you feel that this manuscript is now acceptable for publication, you may indicate that here to bypass the “Comments to the Author” section, enter your conflict of interest statement in the “Confidential to Editor” section, and submit your "Accept" recommendation.

Reviewer #1: All comments have been addressed

Reviewer #2: All comments have been addressed

2. Is the manuscript technically sound, and do the data support the conclusions?

Reviewer #1: Yes

Reviewer #2: Yes

3. Has the statistical analysis been performed appropriately and rigorously? 

Reviewer #1: Yes

Reviewer #2: Yes

4. Have the authors made all data underlying the findings in their manuscript fully available?

Reviewer #1: Yes

Reviewer #2: Yes

5. Is the manuscript presented in an intelligible fashion and written in standard English?

Reviewer #1: Yes

Reviewer #2: Yes

6. Review Comments to the Author

Reviewer #1: The authors have addressed all comments, most of them in a fully satisfactory fashion.

The one point I still think hurts the paper is the claim that the study can be seen as a methodological "breakthrough". It does offer an interesting comparison between behavior in the lab and a remote setting, but these methodologies are both well-known. The paper does provide evidence that behavior seems to be similar when comparing the results from these two settings, but this is just a marginal piece of evidence in the broader picture. For such a paper to constitute a breakthrough, one would need more than a general comparison of behavior across some (important) games. For example, it should be clear to everyone that under some circumstances the full experimental control possible in the lab is desirable and required to test a theory/prediction/behavior. So, a methodological breakthrough could be a clear characterization of questions/games/environments that need more control and when interaction via individual's personal electronic devices is likely sufficient.

Perhaps this is just a semantic issue. But to me the introduction is overall less compelling because of the "breakthrough" phrasing and the reasoning around it. It discounts everything that follows. Why not just say that the paper contributes to a better understanding of the differences between the lab and virtual experiment, and that, for the analyzed games, there are no substantial differences (at least not for the variables the analysis focuses on).

In any case, this is an interesting study with some intriguing findings.

Reviewer #2: (No Response)

7. PLOS authors have the option to publish the peer review history of their article (what does this mean?). If published, this will include your full peer review and any attached files.

Reviewer #1: No

Reviewer #2: No

---

## [Editor Report · Acceptance letter]

20 Apr 2021

PONE-D-20-38803R1 

Conducting large, repeated, multi-game economic experiments using mobile platforms 

Dear Dr. Duffy:

I'm pleased to inform you that your manuscript has been deemed suitable for publication in PLOS ONE. Congratulations! Your manuscript is now with our production department. 

Kind regards, 

on behalf of

Dr. Douglas D. Davis 

Academic Editor

PLOS ONE